# KEPIL: Knowledge-Enhanced Prompt-Image Learning for Prompt-Robust Disease Detection

## Abstract

Vision–language models (VLMs) show promise for clinical decision support in radiology because they enable joint reasoning over radiological images and clinical text, thereby leveraging complementary clinical information. However, radiological findings are long-tailed in practice, leaving some conditions underrepresented and making zero-shot inference essential. Yet current CLIP-style medical VLMs are sensitive to prompt variations and often lack trustworthy external knowledge at inference time, which hinders reliable clinical deployment. We present *KEPIL*, a prompt-robust framework that integrates curated medical knowledge to stabilize zero-shot generalization. KEPIL comprises: (i) *dynamic prompt enrichment* using ontologies with LLM assistance, (ii) a *semantic-aware contrastive loss* aligning embeddings of equivalent prompt variants via a dual-embedding objective, and (iii) *entity-centric report standardization* to yield ontology-aligned representations. Across seven benchmarks, KEPIL achieves state-of-the-art zero-shot/finetuning performance in classification and segmentation; under prompt-variation tests, it improves AUC by $6.37\%$ on *CheXpert* and by $4.11\%$ on average. Ablations and qualitative analyses validate the contributions of enriched prompts and semantic alignment, while attention maps highlight clinically relevant regions. These results suggest that structured knowledge and robust prompt design are key to clinically reliable radiology-facing VLMs. Code will be released at $\star\star\star$.

## 1 Introduction

Vision–language models (VLMs) have become a compelling paradigm for medical AI by enabling joint reasoning over radiological images and accompanying clinical text. Nevertheless, radiological findings exhibit pronounced long-tailed distributions and impose substantial expert-annotation burdens, leaving many conditions underrepresented during training (Zhang & Metaxas, 2024; Holste et al., 2022; Lin et al., 2025). In this context, zero-shot inference is indispensable: for rare diseases, available cases are too limited to support supervised learning. Notably, standardized medical ontologies and radiographic descriptors are often available even for underrepresented conditions, enabling their semantic alignment with well-characterized common diseases. Such knowledge-driven alignment facilitates reliable diagnosis in data-sparse regimes (Zhang et al., 2023; Lin et al., 2025; Rahman et al., 2024), thereby advancing the practical readiness of VLMs for clinical deployment. Despite this potential, their practical deployment remains limited by two key challenges. First, current CLIP-style medical VLMs (Zhang et al., 2022; Huang et al., 2021; Boecking et al., 2022; Wu et al., 2023b) are highly sensitive to variations in textual prompts. Most existing approaches rely on fixed or template-based prompts during training and inference (Zhang et al., 2023; Wu et al., 2023b; Lai et al., 2024), yet even minor changes in phrasing at test time can lead to substantial performance degradation as illustrated in Figure 2. This lack of robustness undermines their reliability in real-world clinical settings, where user inputs are naturally diverse. Second, robust generalization to rare or unseen diseases hinges on dependable external knowledge bases. However, prior work often constructs disease knowledge solely via LLMs, inviting hallucination and inconsistency (Huang et al., 2025; Zhang et al., 2025).

To address these issues, we propose *KEPIL* (Knowledge-Enhanced Prompt Image Learning), a novel framework designed to improve both robustness to prompt variations and transparent zero-shot generalization. We instead ground knowledge in curated resources (e.g., Radiopaedia, UMLS) and use

LLMs only to draft and normalize text under ontology constraints. This ontology-aligned, verifiable knowledge base mitigates hallucination and stabilizes zero-shot inference beyond the training distribution. KEPIL incorporates structured medical knowledge through three key components: (1) dynamic prompt enrichment using medical ontologies and large language models to generate diverse, clinically grounded descriptions; (2) a semantic-aware contrastive loss that encourages consistency among embeddings of prompt variants while separating semantically different prompts; and (3) entity-centric report standardization that reduces sensitivity to linguistic variation. Extensive experiments on seven benchmark datasets demonstrate that KEPIL achieves state-of-the-art performance in zero-shot classification and segmentation, with notable improvements in robustness to prompt variation and enhanced recognition of rare conditions. These findings underscore the critical role of prompt design and knowledge integration in developing clinically reliable VLMs.

## 2 RELATED WORK

Vision-language pretraining (VLP) methods (Yao et al., 2021; Radford et al., 2021; Jia et al., 2021; Wang et al., 2025; Xu et al., 2023) have revolutionized multimodal learning by jointly modeling images and text in recent years. Although models like CLIP (Radford et al., 2021) and MetaCLIP (Xu et al., 2023) have impressive performance on large-scale open-domain tasks, their adaptation to medical imaging remains challenging. Medical images demand finer-grained feature extraction and robust handling of complex clinical language. This section reviews related work from three aspects: medical-specific VLP approaches, decomposition of disease descriptions, and knowledge-enhanced prompting and standardization.

### 2.1 MEDICAL VISION-LANGUAGE PRE-TRAINING

To address the gap between general VLP models and the unique challenges in medical imaging, several studies have adapted the VLP paradigm to the medical domain. Early works such as ConVIRT (Zhang et al., 2022) and GLoRIA (Huang et al., 2021) leverage contrastive learning to align chest radiographs with detailed radiology reports. These methods incorporate both global image–text correspondences and local region-level alignments to capture clinical details. However, due to the complex and specialized nature of clinical language, these approaches often suffer from sensitivity to prompt phrasing and limited generalization. In real-world applications, user inputs may vary substantially, including minor typographical errors, further undermining robustness. To mitigate this, we introduce a semantic-aware contrastive loss that explicitly aligns representations of semantically similar prompts, thereby reducing prompt sensitivity and suggesting improved clinical reliability.

### 2.2 KNOWLEDGE-ENHANCED PRETRAINING

To improve models' comprehension of images, a range of pretraining techniques has been developed to embed domain-specific expertise. These approaches can broadly be categorized into two groups: data structure compatibility (He et al., 2019; Liu et al., 2020) and knowledge supervision (Wang et al., 2021; Zhang et al., 2023; Wang et al., 2022; Wu et al., 2023a). The latter has proven particularly effective in fields like automated medical diagnosis. Knowledge supervision methods for infusing medical expertise into models are further split into model-based and input-based strategies. Model-based approaches focus on directly embedding radiological or diagnostic workflows into the model architecture (Li et al., 2019; Wang et al., 2020; Huang et al., 2020; Tiu et al., 2022). Conversely, input-based methods leverage medical knowledge as supplementary input during computational tasks or as a guide throughout the training phase (Xie et al., 2018; Tan et al., 2019; Zhang et al., 2023; Wu et al., 2023b; Wittmann et al., 2023), a strategy often employed in tasks like generating medical reports. Despite progress in knowledge-integrated pretraining, current methods fail to capture the fine-grained discriminative knowledge critical for diagnostics. Our approach addresses this gap by going beyond ontology-level descriptions to incorporate detailed radiological features during pretraining, yielding more precise diagnostic cues.

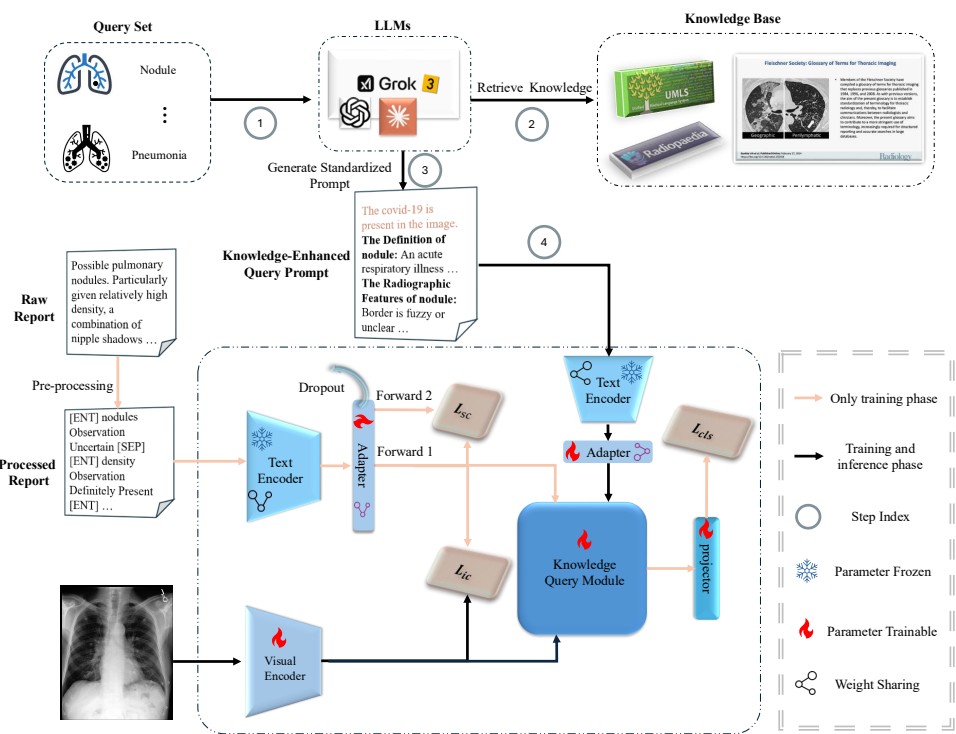

Figure 1: Overview of the proposed KEPIL framework. The model takes three inputs: a chest X-ray image, its corresponding radiology report, and a knowledge-based entity description. During training, the radiology report is parsed to extract anatomical regions, identified pathologies, and the associated confidence levels of those observations. Two forward passes with different dropout configurations are applied to the adapter to compute the semantic-aware contrastive ($\mathcal{L}_{sc}$). An instance contrastive loss ($\mathcal{L}_{ic}$) is then computed between the report text embeddings and the image embeddings. These embeddings, along with knowledge-based entity embeddings, are fed into a description query network to compute the classification loss ($\mathcal{L}_{cls}$). Knowledge enhancement is achieved by using LLMs to extract definitions and descriptions from medical knowledge bases, producing consistent and structured entity descriptions for training.

## 2.3 DECOMPOSITION OF DISEASE DESCRIPTIONS

An emerging line of research addresses the challenge of fine-grained alignment by decomposing disease descriptions into multiple aspects. Approaches such as MAVL (Phan et al., 2024; Jiang et al., 2024; Huy et al., 2025) break down a disease's description into elemental features such as texture, shape, location, and boundary characteristics. This decomposition allows models to associate specific visual cues with corresponding clinical descriptions more precisely. For instance, rather than aligning an image solely with a disease label, these methods link detailed aspects, like the fuzzy boundaries of consolidation or the roundness of a lung nodule, to their respective textual descriptions. Such fine-grained matching enhances zero-shot recognition and improves diagnostic performance, though it also introduces additional complexity in reliably extracting and integrating these detailed descriptors. However, extracting such fine-grained descriptors solely via LLMs risks hallucination and inconsistency. We instead ground descriptor construction in curated external knowledge bases, using LLMs only for ontology-constrained drafting.

## 3 METHOD

Given a dataset $\mathcal{D}$ that comprises $N$ samples, each sample represented by a pair $(\mathbf{x}, \mathbf{t})$, where $\mathbf{x}$ is a radiographic image of size $H \times W$, and $\mathbf{t}$ is its corresponding radiology report. In addition, each sample is accompanied by a collection of definitions and visual depictions for $k$ medically significant entities, which we denote by $\mathbf{d} = \{d_i\}_{i=0}^{k}$. Our goal is to construct a vision-language model $\boldsymbol{\Phi}$ that, in a zero-shot scenario, computes the probability $\hat{\mathbf{y}}$ indicating the presence of a specific finding in a given test radiograph $\mathbf{x}$ with the detailed description of the finding. To achieve this, we propose a strategy comprising the following components as shown in Figure 1: (1) Collect detailed visual descriptions and general definitions of findings from the knowledge base, (2) Design prompt for pretraining and zero-shot inference, (3) Design effective network architectures and inherit knowledge from other powerful pretraining models, and (4) Devise a training protocol to align the visual and text modalities effectively.

**3.1 Knowledge Processing.**

(i) *Pre-processing reports.* Following KAD (Zhang et al., 2023), radiology reports are relatively noisy, and several works have recommended a preprocessing step before their use. For this, we employ RadGraph (Jain et al., 2021), which takes a report ($\mathbf{t}$) as input and recognizes all the radiological entities in it ($e_i | i \in \{1, 2, ...j\}$). It also classifies each of these entities into one of four categories ($s_i$): anatomy (`ANAT`) and observation (`OBS`) to be `Definitely Present`, `Definitely Absent`, or `Uncertain`. The final preprocessed report $\mathbf{e}$ is a concatenation of all the entities and their categories, i.e., $\mathbf{e} = \{e_1, s_1, \texttt{[SEP]}, e_2, s_2, \texttt{[SEP]}, ...e_j, s_j\}$. Additionally, we collect the $M$ most relevant entities in the training dataset into an entity set $\mathcal{E}$, for which the visual descriptions are collected as described below.

(ii) *Collecting and processing visual descriptions.* While prior work has typically utilized standardized medical definitions (e.g., from UMLS) (Silva-Rodriguez et al., 2023; Zhang et al., 2023), radiologists often rely on rich, descriptive visual cues when interpreting imaging findings. To incorporate this aspect, we collect enriched descriptions for entities in $\mathcal{E}$ by merging the UMLS definition with the "radiographic features" section from Radiopaedia (Radiopaedia, 2024) and glossary of terms for thoracic imaging (Hansell et al., 2008) with employing LLMs to standardize descriptions and complete the missing knowledge in the database, ensuring their format and content consistency. Corresponding prompt could be found in supplementary materials

(iii) *Prompt template construction.* We propose a prompt template construction strategy to effectively guide the model in generating structured and semantically rich radiology reports. Given a specific finding, the prompt is constructed by integrating three key components: **(1) Presence statement:** This component explicitly asserts the existence of the finding in the image (e.g., "<finding> is present in the image"), providing an initial observation that anchors the report. **(2) Definition:** Here, a concise definition of the finding is provided using a phrase such as "The Definition of <finding>". This segment offers background information on the pathological or clinical context of the finding. **(3) Radiographic features:** This part details the typical imaging characteristics of the finding, introduced with "The Radiographic Features of <finding>". It includes descriptions of key radiological signs such as border clarity, fluid accumulations, lesion location, opacity types, and other salient features.
By constructing the prompt in this manner, the generated report is designed to capture all essential clinical and radiological information, thereby establishing a clear semantic alignment between image features and textual descriptors.

**3.2 Network Architecture.** Our model consists of two encoding streams: one for images and another shared between the preprocessed report and radiographic descriptions. The image and text features are fused into a unified feature space using a specialized cross-view fusion module. The fused features are then decoded into logits, which are subsequently transformed into probability values. Figure 1 illustrates an overview of KEPIL's architecture.

(i) *Image encoding.* A chest X-ray is encoded by applying a visual backbone $\Phi_{\text{image}}$ to the input image $\mathbf{x}$, yielding a feature map $v = \Phi_{\text{image}}(\mathbf{x}) \in \mathbb{R}^{P \times d}$. In the context of the ViT architecture (Dosovitskiy et al., 2020), the parameter $P$ denotes the number of image patches, while $d$ indicates the feature dimension for each patch. A dedicated learnable `class` token, $v^{cls} \in \mathbb{R}^d$, is used to represent the global encoding of the input image.

**(ii) *Report and knowledge encoding.*** The report $\mathbf{e}$ and its corresponding enriched query prompt $\mathbf{p}$ for detected entities are processed using a knowledge encoder. After tokenizing the report $\mathbf{e}$ and the corresponding query prompt set $\{p_i\}_{i=0}^k$, the tokens are converted into text embeddings. This is expressed as:

$$t_{e^*} = \Phi_{\text{text}}(\mathbf{e}) \in \mathbb{R}^{T \times d}, \quad t_{p^*} = \Phi_{\text{text}}(p_i) \in \mathbb{R}^{T \times d},$$

where $T$ indicates the total number of text tokens, $d$ represents the feature dimensionality. Additionally, the text encoder generates global CLS tokens $t_{e^*}^{cls}$ and $t_{p^*}^{cls}$, which capture the overall semantic content of the report and descriptions, respectively. After adapter processing, $t_{e^*}$ and $t_{p^*}$ become $t_e$ and $t_p$, respectively.

**(iii) *Knowledge-based query for diagnosis.*** Compared to natural images, medical images, such as chest radiographs, contain intricate, fine-grained details critical to specific domains. To enhance the model's ability to discern challenging cases, we introduce a cross-attention module, termed the *Knowledge Query Module* (KQM), for token-level alignment. The KQM aligns radiograph visual tokens with text tokens from the report and prompt. It employs multiple cross-attention layers, where projections of either report or prompt tokens form key and value pairs, queried by visual embeddings associated with local patches. In the KQM, the prompt embedding $t_p^{cls} \in \mathbb{R}^{S \times d}$ is projected into queries $Q$, where $S$ represents the query set size, while the patch-wise visual embeddings $v \in \mathbb{R}^{P \times d}$ are encoded into keys $K$ and values $V$. The enriched embedding obtained through cross-attention querying is then computed as $t_{attn}^{cls} = \text{softmax}\left(\frac{QK^\top}{\sqrt{d}}\right)V$, where $t_{attn}^{cls} \in \mathbb{R}^{S \times d}$.

### 3.3 Training Objectives.

**(i) *Semantic-aware contrastive learning.*** To enhance the expressive power of the frozen text encoder without modifying its pretrained weights, we introduce an adapter module. This module aims to improve the discriminative capability of raw text embeddings and is trained using a contrastive loss ($\mathcal{L}_{sc}$) that maximizes consistency between the original and adapter-enhanced representations. Specifically, for a radiograph-report pair $(\mathbf{x}, \mathbf{t})$, we first extract the frozen text embedding from the CLS token, denoted as $t_{e^*}^{cls}$. The adapter then transforms this embedding as follows: $t_e^{cls} = \mathcal{A}(t_{e^*}^{cls})$, where $\mathcal{A}$ denotes the adapter function. We repeat this process to obtain $t_e^{cls^2}$. Due to the dropout in the adapter, $t_e^{cls}$ and $t_e^{cls^2}$ are two perturbed views. The adapter is utilized to learn the consistency from two views of the embedding and the contrastive loss $\mathcal{L}_{sc}$ is applied to maximize the semantic consistency between $t_e^{cls}$ and $t_e^{cls^2}$ while ensuring they are discriminated against embeddings from other samples in a training batch $B$. Formally, the loss is defined as:

$$\mathcal{L}_{sc}(t_e^{cls}, t_e^{cls^2}) = -\log \frac{\exp\left(t_e^{cls}, t_e^{cls^2}\right)/\tau)}{\sum_{t_e'^{cls} \in B} \exp\left(s(t_e^{cls}, t_e'^{cls})/\tau\right)} \tag{1}$$

where $s(\cdot, \cdot)$ represents the cosine similarity and $\tau$ is the temperature parameter. $B$ represents the batch.

**(ii) *Instance level alignment of radiograph-report.*** The global alignment between an image and its corresponding radiology report is accomplished using the standard image-text contrastive loss, denoted as $\mathcal{L}_{ic}$. This loss function maximizes the mutual information between the image and report representations. In our approach, we specifically utilize the report embedding because it effectively captures the overall context of the image. Given a triplet consisting of a radiograph, its report, and a description $(\mathbf{x}, \mathbf{e}, \mathbf{d})$, our goal is to enhance the similarity between the embedded image and report features, with a particular focus on their CLS token representations. Formally, within a training batch $B$, the loss is defined as:

$$\mathcal{L}_{ic}(v^{cls}, t_e^{cls}) = -\log \frac{\exp\left(s(v^{cls}, t_e^{cls})/\tau\right)}{\sum_{t_e'^{cls} \in B} \exp\left(s(v^{cls}, t_e'^{cls})/\tau\right)} \tag{2}$$

**(iii) *Overall loss.*** The overall training objective integrates the contrastive report embedding loss, the instance-level image-text contrastive loss (Eq. 2) and the binary classification loss. Additionally, we employ a binary cross-entropy loss ($\mathcal{L}_{cls}$), which is applied only when queries from $\mathcal{E}$ explicitly appear or are explicitly absent in the report. In cases where the presence of findings is uncertain, this loss is not computed. Specifically, the names of the entities are encoded by $\Phi_{text}$ and refined

through the knowledge-retrieval module to generate the output logits. The total loss to be minimized is: $\mathcal{L} = \lambda_1 \mathcal{L}_{cls} + \lambda_2 \mathcal{L}_{ic} + \lambda_3 \mathcal{L}_{sc}$, where $\lambda$ regulates the contribution of radiographic descriptions.

## 4 EXPERIMENT SETTINGS

### 4.1 PRE-TRAINING AND DOWNSTREAM DATASET

We evaluate our approach on seven publicly available chest-X-ray datasets covering classification, segmentation, and grounding tasks: **Classification:** CheXpert (Irvin et al., 2019), ChestXray-14 (Wang et al., 2017), PadChest (Bustos et al., 2020), RSNA Pneumonia (Wu et al., 2024), SIIM-ACR (Anna Zawacki et al., 2019), COVIDx CXR-2 (Pavlova et al., 2022) and LIDC-IDRI dataset (Armato III et al., 2011); **Segmentation:** RSNA Pneumonia (Wu et al., 2024), SIIM-ACR (Anna Zawacki et al., 2019) and COVID Rural (Desai et al., 2020); **Grounding:** RSNA Pneumonia (Wu et al., 2024) and COVID Rural (Desai et al., 2020). Details on dataset splits and preprocessing procedures are provided in the supplementary materials.

### 4.2 IMPLEMENTATION DETAILS

The text encoder is initialized with BioClinicalMPBERT (Lai et al., 2024) weights during pretraining and remains frozen. The image encoder is ViT-B/16 which utilizes M3AE for pretraining only on the MIMIC dataset (Lai et al., 2024). For the adapters, we use light-weighted two-layer MLPs with dropout set to 0.5 by default at the report branch. The adapter, projector, visual encoder and KQM module are trainable. The ChatGPT-4o is utilized for prompt standardization and completion as it outperforms the other close/open-resource LLMs in the description completion task as shown in supplementary materials. For generating prompt variants, we include not only rephrasings but also realistic errors such as typos, omissions, and incorrect punctuation. The pretraining is conducted on two H100 GPUs. For finetuning, following (Phan et al., 2024), we use ViT-B (Dosovitskiy et al., 2020) for classification tasks and adopt the TransUNet (Chen et al., 2021) type architecture for segmentation tasks, initializing these models with our pre-trained visual encoder weights. The finetuning experiments are conducted on four RTX4090 GPUs. Additional hyperparameter details are provided in the supplementary material. We adopt standard evaluation metrics for classification and segmentation tasks, consistent with previous work (Wu et al., 2023b; Boecking et al., 2022). We report the Area Under the Curve (AUC), F1 score, and accuracy for classification. We use Dice score, Intersection over Union (IoU), and pixel-wise accuracy for segmentation. All metrics are reported as percentages for clarity and consistency.

Table 1: Comparisons with SOTA image-text pretraining models under zero-shot classification for base diseases that have been seen during pre-training across five datasets. The best results are bolded, while the second-best are underlined.

| Dataset | CheXpert | | | ChestXray-14 | | | PadChest-seen | | | RSNA Pneumonia | | | SIIM-ACR | | |
|---|---|---|---|---|---|---|---|---|---|---|---|---|---|---|---|
| Method | AUC↑ | F1↑ | ACC↑ | AUC↑ | F1↑ | ACC↑ | AUC↑ | F1↑ | ACC↑ | AUC↑ | F1↑ | ACC↑ | AUC↑ | F1↑ | ACC↑ |
| ConVIRT (Zhang et al., 2022) | 52.10 | 35.61 | 57.43 | 53.15 | 12.38 | 57.88 | 63.72 | 14.56 | 73.47 | 79.21 | 55.67 | 75.08 | 64.25 | 42.87 | 53.42 |
| GLoRIA (Huang et al., 2021) | 54.84 | 37.86 | 60.70 | 55.92 | 14.20 | 59.47 | 64.09 | 14.83 | 73.86 | 70.37 | 48.19 | 70.54 | 54.71 | 40.39 | 47.15 |
| BioViL (Boecking et al., 2022) | 60.01 | 42.10 | 66.13 | 57.82 | 15.64 | 61.33 | 60.35 | 10.63 | 70.48 | 84.12 | 54.59 | 74.43 | 70.28 | 46.45 | 68.22 |
| BioViL-T (Boecking et al., 2022) | 70.93 | 47.21 | 69.96 | 60.43 | 17.29 | 62.12 | 65.78 | 15.37 | 77.52 | 86.03 | 62.56 | 80.04 | 75.56 | 60.18 | 73.72 |
| CheXzero (Tiu et al., 2022) | 87.90 | 61.90 | 81.17 | 66.99 | 19.95 | 65.38 | 73.24 | 19.53 | 83.49 | 83.13 | 61.49 | 78.34 | 84.60 | 65.97 | 77.34 |
| MedKLIP (Wu et al., 2023b) | 87.97 | 63.67 | 84.32 | 72.33 | 24.18 | 79.40 | 77.87 | 26.63 | 92.44 | 86.57 | 63.28 | 79.97 | 89.79 | 72.73 | 83.99 |
| KAD (Zhang et al., 2023) | 89.23 | 63.25 | 86.25 | 76.49 | 29.98 | 80.49 | 80.94 | 58.23 | 92.12 | 85.32 | 87.09 | 81.80 | 87.39 | 68.84 | 81.73 |
| MAVL (Phan et al., 2024) | 90.13 | 65.47 | 86.44 | 73.57 | 26.25 | 82.77 | 78.79 | 28.48 | 92.56 | 86.91 | 63.41 | 82.42 | 92.04 | 77.95 | 87.14 |
| CARZero (Lai et al., 2024) | **92.38** | 40.20 | 86.93 | 79.64 | 31.41 | 79.60 | 83.97 | 25.08 | 91.46 | 80.28 | 28.28 | 41.52 | 90.90 | 75.96 | 85.03 |
| **KEPIL (Ours)** | 91.21 | 65.56 | 87.88 | 80.95 | 34.74 | 83.23 | 85.32 | 59.79 | 93.16 | 89.76 | 90.23 | 86.24 | 93.02 | 79.22 | 87.46 |

## 5 RESULTS

### 5.1 ZERO-SHOT EVALUATION

**(i) *Classification for seen diseases.*** Table 1 presents the zero-shot classification results for *seen* diseases. *Seen* diseases are those that appear in the MIMIC dataset, which the model is exposed to during pretraining. Compared to other vision-language models (VLMs) pretrained on the MIMIC dataset, our proposed method, KEPIL, consistently achieves the highest performance in terms of

Table 2: Zero-shot performance on unseen and rare diseases as well as on an unseen modality. COVID-19 CXR-2 and PadChest-unseen include categories absent from pretraining; PadChest-rare includes categories with few positives; LIDC-IDRI evaluates transfer from CXR to CT. The model maintains strong zero-shot performance across all settings, evidencing robust generalization to novel categories, scarce samples, and modality shift.

| Dataset Method | Covid-19 CXR-2 (CXR) AUC↑ | ACC↑ | PadChest-unseen (CXR) AUC↑ | ACC↑ | PadChest-rare (CXR) AUC↑ | ACC↑ | LIDC-IDRI (CT) AUC↑ |
|---|---|---|---|---|---|---|---|
| ConVIRT | 62.78 | 63.84 | 51.17 | 61.51 | 50.37 | 60.17 | - |
| GLoRIA | 64.52 | 60.21 | 49.96 | 60.95 | 48.25 | 58.49 | - |
| BioViL | 61.40 | 58.20 | 57.95 | 62.50 | 52.82 | 60.60 | - |
| BioViL-T | 62.43 | 57.65 | 58.94 | 68.56 | 57.44 | 65.38 | - |
| CheXzero | 73.13 | 71.45 | 66.70 | 81.19 | 65.08 | 81.17 | - |
| MedKLIP | 76.28 | 71.96 | 60.31 | 76.69 | 59.75 | 77.84 | 45.56 |
| KAD | 74.03 | 72.42 | 73.08 | 83.23 | 72.21 | 86.54 | 57.75 |
| MAVL | **83.86** | **78.07** | 70.42 | 84.00 | 70.06 | 84.64 | 41.81 |
| CARZero | 75.53 | 68.55 | 78.95 | 85.65 | 77.79 | 91.77 | 60.57 |
| **KEPIL (Ours)** | 79.55 | 78.12 | 79.05 | 94.02 | 78.75 | 95.47 | 66.65 |

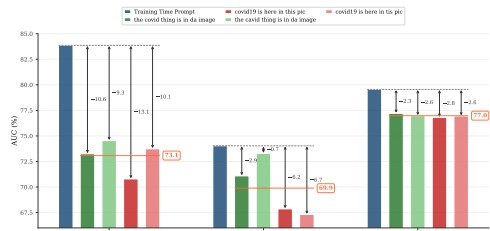

Figure 2: Zero-shot Performance of MAVL, KAD and our KEPIL under prompt perturbations on COVID-19 CXR-2 dataset. Both previous SOTA models exhibit substantial sensitivity to prompt variations, where even a single-character typo can induce an AUC change up to 3.0%.

Table 3: Results of different VLP models under finetuning classification and segmentation with different data portions (10% and 100%). AUC scores and Dice scores are reported for both classification and segmentation tasks.

| Method | Classification | | | | | | | | Segmentation | | | | | |
|---|---|---|---|---|---|---|---|---|---|---|---|---|---|---|
| | RSNA Pneumonia | | SIIM-ACR | | Covid-19 CXR-2 | | ChestXray-14 | | RSNA Pneumonia | | Covid-19 Rural | | SIIM-ACR | |
| Data portion | 10% | 100% | 10% | 100% | 10% | 100% | 10% | 100% | 10% | 100% | 10% | 100% | 10% | 100% |
| Scratch | 83.31 | 87.12 | 76.18 | 87.48 | 93.65 | 98.86 | 56.27 | 67.03 | 58.42 | 69.75 | 25.97 | 37.83 | 33.23 | 74.47 |
| ConVIRT (Zhang et al., 2022) | 85.42 | 87.64 | 80.41 | 91.67 | 97.74 | 99.70 | 72.53 | 79.13 | 63.94 | 71.87 | 30.79 | 42.71 | 61.21 | 73.52 |
| GLoRIA (Huang et al., 2021) | 85.59 | 87.83 | 86.20 | 91.89 | 97.18 | 99.54 | 72.87 | 79.92 | 67.71 | 72.06 | 31.20 | 43.85 | 57.78 | 76.94 |
| BioViL (Boecking et al., 2022) | 86.04 | 88.29 | 79.45 | 88.05 | 98.39 | 99.68 | 72.94 | 80.16 | 68.72 | 73.64 | 37.75 | 47.34 | 69.98 | 78.49 |
| MedKLIP (Wu et al., 2023b) | 87.14 | 88.58 | 89.91 | 93.01 | 98.77 | 99.77 | 74.02 | 80.79 | 70.24 | 73.88 | 39.28 | 48.65 | 72.10 | 79.37 |
| KAD (Zhang et al., 2023) | 82.98 | 89.70 | 90.81 | 92.22 | 98.64 | 99.86 | 71.25 | 80.35 | 72.20 | 76.16 | 40.00 | 43.39 | - | - |
| MAVL (Phan et al., 2024) | 87.90 | 88.94 | 93.00 | 94.48 | 99.15 | 99.90 | 80.02 | 83.32 | 73.51 | 76.97 | 40.71 | 50.25 | - | - |
| KEPIL (Ours) | 89.12±0.12 | 93.06±0.40 | 94.92±0.21 | 95.15±0.26 | 99.82±0.01 | 99.98±0.01 | 80.74±0.11 | 83.45±0.19 | 76.72±0.95 | 78.04±0.63 | 48.27±0.93 | 53.88±1.27 | 73.25±0.22 | 80.11±0.43 |

both AUC and ACC across all five datasets. For instance, KEPIL outperforms the second-best method, CARZero, by 1.31% on the ChestXray-14 dataset and by 2.85% on the RSNA Pneumonia dataset in terms of AUC. Additionally, KEPIL achieves the best or second-best performance on F1 scores. These results demonstrate the model's robustness and effectiveness in zero-shot disease detection under significant domain shifts.

**(ii)** *Classification for unseen and rare diseases.* Table 2 reports zero-shot classification on diseases absent from pretraining ("unseen") and on diseases sparsely observed during pretraining ("rare"). On the two unseen datasets, COVID-19 CXR-2 and PadChest-unseen, our proposed method, KEPIL, achieves either the best or second-best performance when compared to other vision-language models. Specifically, KEPIL demonstrates comparable performance to MAVL on the COVID-19 CXR-2 dataset and achieves notable improvements of 8.37% in ACC on the PadChest-unseen dataset over CARZero. For rare diseases, which are included during pretraining but with very few positive sample counts, KEPIL significantly improves ACC by 3.70% over the second-best performances. These results underscore KEPIL's strong capability to generalize to novel or low-resource disease categories, further highlighting its potential in real-world clinical applications where data scarcity and domain shifts are common. Moreover, we further evaluated KEPIL's cross-modality generalization on CT slices from the LIDC-IDRI dataset, where it achieved a leading AUC performance that surpassed the second-best method by 6.08%.

**(iii)** *Robustness to Semantic Variations of Prompts.* Figure 4 reports zero-shot AUCs using query prompts generated by Grok3, ChatGPT-4o, and Claude Sonnet 4. Across both in-domain and out-of-domain datasets, KEPIL consistently surpasses other SOTA VLMs, demonstrating strong robustness to prompt variability—including realistic errors such as typos, omissions, and incorrect punctuation/formatting. Leveraging this robustness, KEPIL operates with minimal specification: given only a finding name, it automatically composes comprehensive, clinically grounded queries, thereby streamlining prompt creation and improving diagnostic accuracy, transparency, and efficiency.

## 5.2 FINE-TUNING EVALUATION

**(i)** *Finetuning performance on classification tasks* As shown in Table 3, KEPIL achieves competitive fine-tuning performance across all classification benchmarks, outperforming MAVL, KAD, MedKLIP, and CheXZero under both 10% and full-data settings. With only 10% samples, KEPIL

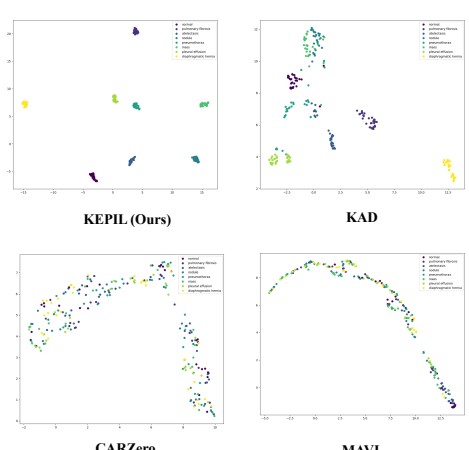

**KEPIL (Ours)**     **KAD**

**CARZero**     **MAVL**

Figure 3: UMAP projection of embeddings generated by multiple clinical text encoders from 30 prompt variations per disease term. KEPIL exhibits tighter clusters, demonstrating superior robustness to prompt variations.

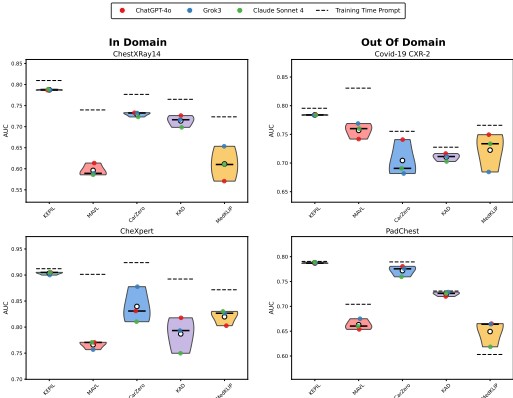

Figure 4: Evaluation of the impact of prompts from ChatGPT-4o, Grok-3, and Claude Sonnet 4 (including syntactic paraphrases, typos, omissions, punctuation variants) on chest X-ray model performance relative to the original training prompt. KEPIL demonstrates smaller performance declines compared to other models.

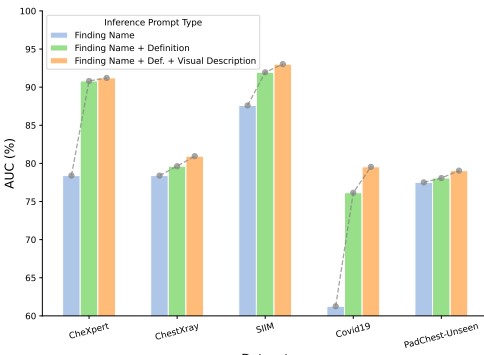

Figure 5: KEPIL's performance across incremental different inference input types. Performance improves with increasingly rich textual input, highlighting the value of more informative prompts.

Table 4: Ablation study of the effects of enriched prompts, the inclusion and placement of $\mathcal{L}_{sc}$, and dropout rate on zero-shot inference performance across the *ChestXRay14* and *CheXpert* datasets.

| Ablation | ChestXRay14 | CheXpert |
|---|---|---|
| *Effect of Enriched Prompt and $\mathcal{L}_{sc}$* | | |
| Base ($\mathcal{L}_{cls} + \mathcal{L}_{ic}$) | 76.89 | 87.07 |
| + Enriched Prompt (EP) | 79.45 | 90.47 |
| + EP + $\mathcal{L}_{sc}$ | **80.95** | **91.21** |
| *Effect of Dropout Rate in $\mathcal{L}_{sc}$* | | |
| Dropout = 0.3 | 80.11 | 90.98 |
| Dropout = 0.4 | 80.29 | 91.41 |
| Dropout = 0.5 | **80.95** | **91.21** |
| Dropout = 0.6 | 80.24 | 91.24 |
| *Effect of the Position of $\mathcal{L}_{sc}$* | | |
| Report branch | **80.95** | **91.21** |
| Prompt branch | 79.72 | 90.52 |
| Report branch + Prompt branch | 80.19 | 91.15 |

surpasses MAVL's performance finetuned with 100% data on the RSNA dataset and SIIM-ACR dataset. These improvements address that the stable text–image alignment learned during pre-training produce more transferable representations, allowing KEPIL to adapt efficiently even with limited labels.

**(ii)** *Finetuning performance on segmentation tasks* KEPIL attains the best or second-best performance across RSNA Pneumonia, SIIM-ACR, and COVID Rural in the segmentation setting as shown in Table 3. In particular, KEPIL surpasses the second-best method, MAVL, by a notable margin of 3.63% on the Covid-19 Rural segmentation dataset. The trend is consistent with its zero-shot grounding results in Table 5, where KEPIL produces more accurate attention on the same dataset, indicating that KEPIL has learnt strong spatial priors knowledge before supervision. Incorporating these priors enables more effective refinement of lesion regions during fine-tuning, yielding consistently superior segmentation performance across datasets.

Table 5: Comparison with SOTA VLMs on zero-shot region grounding for Pneumonia and Covid-19. Our KEPIL achieves the best performance on 5 of 6 metrics across two datasets

| Method | RSNA Pneumonia | | | Covid-19 Rural | | |
|---|---|---|---|---|---|---|
| | IoU↑ | Dice↑ | ACC↑ | IoU↑ | Dice↑ | ACC↑ |
| GLoRIA (Huang et al., 2021) | 21.82 | 34.68 | 75.14 | 8.18 | 12.49 | 66.73 |
| BioViL (Bannur et al., 2023) | 30.29 | 43.86 | 82.15 | 11.52 | 15.77 | 70.84 |
| MedKLIP (Wu et al., 2023b) | 34.41 | 49.23 | 86.90 | 20.88 | 32.38 | 76.23 |
| KAD (Wu et al., 2023b) | 34.23 | 49.77 | 87.40 | 21.24 | 34.62 | 78.53 |
| MAVL (Phan et al., 2024) | 34.72 | 50.04 | 88.53 | 21.97 | 34.11 | **84.29** |
| KEPIL (ours) | **35.20** | **50.79** | **89.51** | **24.80** | **36.79** | 80.87 |

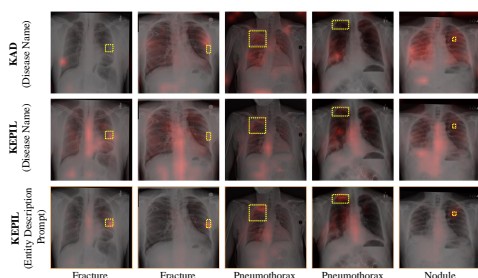

Figure 6: Comparison of attention maps generated by KAD and KEPIL with varying prompts on the ChestX-Det images.

## 5.3 ABLATIONS

(**i**) *Richer Textual Inputs Enhance Diagnostic Performance.* We investigated the impact of varying levels of textual detail on KEPIL's performance. As shown in Figure 5, model performance consistently improves with richer textual inputs. Specifically, using only finding names (e.g., "Atelectasis") results in the lowest AUC scores, while supplementing the names with definitions from the UMLS knowledge base leads to notable gains. The highest performance is achieved when definitions are further enriched with detailed descriptions, highlighting the model's ability to establish a connection between diagnosis and finding-related features. This finding underscores the importance of incorporating rich and structured textual context to enhance the model's robustness and classification accuracy.

(**ii**) *Enriched Prompt and $\mathcal{L}_{sc}$ trigger Significant Performance Gains.* We conduct an ablation study to assess the contribution of each component in the KEPIL framework by progressively adding the Enriched Prompt (EP) and $\mathcal{L}_{sc}$ to the base optimization strategy ($\mathcal{L}_{cls} + \mathcal{L}_{ic}$). As shown in Table 4, incorporating enriched prompts significantly improves zero-shot classification performance, boosting the AUC from 76.89% to 79.45% on ChestXray14 and from 87.07% to 90.47% on CheXpert. Adding $\mathcal{L}_{sc}$ with a dropout rate of 0.5 further enhances performance, particularly on ChestXray14, reaching the highest AUC of 80.95%. Experiments with dropout rates between 0.3 and 0.6 show 0.5 to be optimal. We also evaluated where to apply $\mathcal{L}_{sc}$ and found that placing it in the report branch yields the greatest improvement. Overall, the complete KEPIL model achieves a performance gain of over 4% in AUC compared to the base setup across both datasets, demonstrating the effectiveness of enriched prompts and the proposed $\mathcal{L}_{sc}$.

## 5.4 QUALITATIVE ANALYSIS

In Figure 6, we visualize attention maps generated by KEPIL and KAD on ChestX-Det images, annotated with radiologist-drawn bounding boxes. KAD only finds names as prompts and produces broader and less precise attention regions. In contrast, KEPIL, predominantly when guided by enriched textual prompts that include full-length clinical descriptions, generates attention maps that closely align with annotated pathological areas, particularly for localized findings such as fractures, pneumothorax, and nodules. This comparison demonstrates that incorporating richer semantic information enables the model to focus on clinically relevant areas better, emphasizing the advantage of KEPIL's prompt-enhanced design.

Figure 3 visualizes UMAP projections of text embeddings derived from 30 prompt variants across multiple text encoders. Methods such as KAD and MAVL typically keep their text encoder frozen for simplicity. However, this leads to less discriminative embeddings and increased sensitivity to semantic variations in zero-shot settings. In contrast, our learned encoder produces more distinct clusters for related concepts, demonstrating improved robustness and reduced vulnerability to nuanced language shifts.

## 6 CONCLUSION

We introduce KEPIL, a robust knowledge-enhanced vision-language framework designed to address the critical limitations of prompt sensitivity and poor generalization in open-vocabulary disease detection for radiological imaging. Unlike prior VLMs that rely on fixed or template-based prompts, KEPIL integrates medical ontologies, radiographic descriptors, and entity-level representations into the vision-language pipeline. KEPIL establishes a semantically aligned and clinically grounded interface between text and image modalities with a semantic-aware contrastive learning objective and a prompt standardization protocol, leading to enhanced stability under prompt perturbations and improved generalization to rare and unseen diseases.

## REPRODUCIBILITY STATEMENT

The main training code is provided in the Supplementary Materials. All datasets are publicly accessible, and their splits follow the official split. Training parameters and implementation details are provided in the Implementation Details section and the Supplementary Materials. Additionally, the prompts used to generate the findings-related descriptions are included in the Supplementary Materials.

## ETHICS STATEMENT

We have read the ICLR Code of Ethics (https://iclr.cc/public/CodeOfEthics) and ensured that this work adheres to it.

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
