# OpenReview forum: "KEPIL: Knowledge-Enhanced Prompt-Image Learning for Prompt-Robust Disease Detection"
_ICLR.cc/2026/Conference — Submitted to ICLR 2026_

### Official Review · Reviewer_b7Jv · 2025-10-27

**Soundness:** 3
**Presentation:** 2
**Contribution:** 2
**Rating:** 2
**Confidence:** 5

**Summary:**

This paper proposes KEPIL, a knowledge-enhanced vision-language framework aiming to improve prompt robustness and zero-shot generalization in medical imaging tasks. It integrates curated ontologies (UMLS, Radiopaedia) with LLM-generated descriptions, introduces a semantic-aware contrastive loss (Lsc) to stabilize embeddings across prompt variants, and standardizes radiology reports via entity-centric preprocessing. Experiments on seven public chest X-ray datasets show improved zero-shot and finetuning performance compared to CLIP-style and medical-specific baselines.

**Strengths:**

- Addresses a practically relevant issue: prompt sensitivity and lack of knowledge grounding in medical VLMs.

- Combines knowledge curation and contrastive learning in an interpretable manner.

- Provides comprehensive experiments with both quantitative and qualitative analyses.

- Demonstrates cross-modality transfer and prompt perturbation robustness, which are meaningful for clinical deployment.

**Weaknesses:**

- The work is largely incremental and engineering-oriented. Its main components—ontology-guided prompt design, adapter alignment, and cross-attention fusion—closely follow prior work such as KAD, MAVL, and MedKLIP. The proposed “semantic-aware contrastive loss” is a simple consistency objective, and the “Knowledge Query Module” mainly reuses standard cross-attention. Overall, the paper does not introduce substantial novelty or new conceptual understanding of prompt robustness or knowledge integration.

- The paper also relies heavily on ChatGPT-4o for generating and refining prompts. While this provides flexibility, it introduces potential issues with factual accuracy and consistency, as large language models can produce hallucinated or unstable medical text. The paper does not include verification or expert validation of these outputs. Since the generated text directly affects training, it is unclear whether the reported gains stem from true model robustness or from uncontrolled variations in GPT-generated data.

- The experimental analysis lacks statistical rigor. The paper reports no standard deviations, repeated trials, or significance testing. The reported 1–3% performance gains may fall within normal variance. In addition, the prompt robustness experiments mainly test minor typos rather than diverse or semantically rephrased prompts, providing limited evidence of genuine robustness.

- The claims of clinical generalization appear overstated. All experiments are conducted on public datasets rather than real-world or prospective clinical data. The CXR-to-CT transfer is a simplified setting that does not demonstrate true cross-modality adaptation. Without human or expert validation, the claim of “trustworthy clinical deployment” is not sufficiently supported.

**Questions:**

Please refer to the Weaknesses section.

**I am willing to raise my score according to the rebuttal.**

---

> ### Author Response · Authors · 2025-11-19
>
> # Response to Reviewer b7Jv
>
> We sincerely thank the reviewer for the comprehensive and thoughtful evaluation, as well as the willingness to reconsider the rating. We appreciate the recognition of the practical relevance of addressing prompt sensitivity and knowledge grounding in medical VLMs. At the same time, we take the concerns regarding novelty, LLM reliability, statistical rigor, and clinical claims very seriously. These comments are highly constructive and have guided substantial clarifications and additional experiments in our rebuttal, which we hope effectively address the reviewer’s points.
>
> ### W1 (method)
>
> Our work is indeed built upon prior vision–language models, but the focus is fundamentally different. KEPIL specifically targets **prompt robustness**, an aspect largely overlooked in earlier studies despite its practical importance in real clinical workflows. Our comparisons with previous methods consistently demonstrate that KEPIL provides substantially stronger robustness under real-world prompt variations. Furthermore, the proposed semantic-aware contrastive learning loss is indeed simple yet effective. By relying on dropout-based perturbations, it enforces invariance to superficial linguistic variations while keeping both augmented views strictly within the clinical language manifold. This ensures that the model improves prompt robustness without drifting toward off-manifold natural-language patterns that harm clinical generalization.
>
> ### W2 (Text quality)
>
> We agree that the quality of knowledge is very important. Thanks for the suggestion. We have conducted a double-blind experiment with a radiologist to evaluate the quality of GPT-4o-augmented knowledge for diseases **with (A)** and **without (B)** available raw-text descriptions from Radiopaedia.
>
> **(A) With raw-text (format standardization only):**
> We only use the LLM for format standardization without altering the underlying clinical content. We invited a radiologist to perform a double-blind clinical readability assessment on 10 randomly selected entities [pleural effusion, atelectasis, consolidation, pneumonia, fracture, nodule, hyperinflate, collapse, emphysema, fibrosis].
>
> Scoring rubric:
> - **1** — unclear, disorganized, or difficult to interpret
> - **2** — somewhat readable but awkward or inconsistent
> - **3** — acceptable but could be smoother
> - **4** — clear and well-structured
> - **5** — highly readable, polished, and professional
>
> | Source     | Scores                        | Avg |
> |------------|-------------------------------|-----|
> | Raw text   | [3,5,5,5,5,4,3,4,5,3]          | 4.2 |
> | GPT-4o     | [5,5,5,2,3,5,5,5,4,5]          | 4.4 |
>
> This indicates that LLM-based standardization overall improves clinical readability without modifying core medical content.
>
> **(B) Without raw-text (generation from scratch):**
> We note that in the supplementary materials section of the original submission, we compare the LLM-augmented knowledge with the Radiology ontology knowledge (Radiopedia - Experts generated), the GPT-4o achieves the best performance across metrics like F1-RadGraph. We further compared GPT-4o-generated descriptions with raw-text using an expert accuracy rubric:
>
> - **1** — major factual errors or incorrect radiographic statements
> - **2** — multiple inaccuracies or misleading claims
> - **3** — mostly accurate with minor imprecision
> - **4** — accurate with only small refinements needed
> - **5** — fully accurate and radiologically precise
>
> | Source            | Scores                        | Avg |
> |-------------------|-------------------------------|-----|
> | Raw text (Radiopaedia) | [4,5,5,5,5,4,5,5,5,5]    | 4.8 |
> | GPT-4o            | [5,5,5,4,2,5,5,5,3,5]         | 4.4 |
>
> As expected, GPT-4o is slightly less accurate than expert raw text. However, an average of **4.4** still lies between “accurate with small refinements needed” and “fully accurate,” confirming the generated descriptions remain clinically usable.
>
> ### W3 (statistical rigor)
>
> Following prior work, we first select the best zero-shot checkpoint on the validation set and use this fixed model for all downstream finetuning experiments. This ensures that the supervised results are strictly comparable to the zero-shot setting and originate from the same pre-trained state. No variability is generated as it is a deterministic procedure.
>
> Following previous benchmarks, our finetuning results are the average performance over three runs using random seeds 42, 43, and 44. Based on the reviewer’s recommendation, **we have incorporated standard deviations in the updated draft**. Unfortunately, the official reported performance of baselines (public released) does not include standard deviations.

---

> ### Author Response · Authors · 2025-11-19
>
> ### W4 (claims of clinical generalization)
>
> We agree that public datasets typically do not fully characterize a clinical setting. However, in order to demonstrate the benefits of the proposed method, we rely on them to benchmark against previous methods.
>
> In the experiment, we attempted to approximate clinical distributional variability by evaluating KEPIL in strict zero-shot settings across a broad range of publicly available datasets with differing sources and acquisition characteristics. These unseen-distribution evaluations, together with an extreme CXR-to-CT stress test, were designed to mimic the heterogeneity encountered in real clinical environments. We will clarify this in the paper and note that real-world deployment studies are an important direction for future work. Regarding expert validation, we have conducted a double-blind experiment to verify the quality of the knowledge used for pretraining, where the knowledge proved to be of good quality.

---

> ### Comment · Reviewer_b7Jv · 2025-11-24
>
> Thank you for the detailed response and additional experiments. While I appreciate the authors’ effort in clarifying some points, my main concerns remain largely unresolved. First, the contribution of the work still appears primarily engineering-oriented, without providing new insights or findings that would align with the expectations of ICLR. Second, the clinical trustworthiness and deployment claims are still insufficiently supported. The rebuttal includes radiologist verification on only 10 examples and does not provide qualitative or quantitative evaluation in any real clinical setting, even at a small scale. As a result, the clinical conclusions remain overstated relative to the evidence provided. I encourage the authors to further deepen the analysis of the method and conduct more comprehensive, clinically grounded evaluations for a future submission, either to a conference or to a journal with a broader application focus.

---

> > ### Author Response · Authors · 2025-11-26
> >
> > We thank the reviewer for your comments. We appreciate the opportunity to clarify the purpose of our qualitative validation, the methodological novelty of our approach, and how our evaluation framework aligns with the standard scope of this venue.
> >
> > **W1 & Q1 (“Clinical evaluation on only 10 cases is not enough”)**
> > We respectfully clarify a misunderstanding regarding the purpose of the 10-case evaluation. This specific experiment was conducted in direct response to your previous request to verify the factual accuracy of the LLM-generated prompts and check for hallucinations. This human evaluation was not intended to measure the model’s diagnostic accuracy. It was a strict quality control step to verify the textual correctness of the generated definitions. For such a semantic audit, a double-blind expert review of finding descriptions (10 anatomical/pathological categories) is a standard methodology to ensure the input text is hallucination-free (e.g., TRIPOD-LLM [Gallifant J, Afshar M, Ameen S, et al. The TRIPOD-LLM reporting guideline for studies using large language models[J]. Nature medicine, 2025, 31(1): 60-69.]). The actual diagnostic performance of KEPIL was evaluated on seven large-scale public benchmarks (including CheXpert, ChestX-ray14, RSNA, and external validation on PadChest and COVID-19 datasets) comprising a large number of images (comprising a large number of images over 500,000 Xrays). The 10 finding descriptions were strictly for the text-quality assurance you requested, while the model itself was rigorously tested on standard large-scale test sets.
> >
> > **W2 (“Engineered approach, no new insights”)**
> > We respectfully disagree that our work lacks novelty. As insight, we identify a critical, often overlooked failure mode in current medical VLMs: prompt sensitivity. We show that existing SOTA models suffer significant performance drops (e.g., AUC drops from ~92% to ~62%) when subjected to realistic clinical prompt variations like synonyms or jargon. As contribution, KEPIL addresses this via a novel Semantic-Aware Contrastive Loss . This is not merely engineering, but a specific representation learning contribution, regularizing the manifold of prompt variations, which results in SOTA zero-shot performance and has been recognized as a "significant performance gain" by other reviewers.
> >
> > **W3 & Q3 (“Need for a real clinical setting evaluation”)**
> > While we agree that prospective clinical trials are the gold standard for medical devices, we respectfully submit that a "real clinical setting evaluation" (e.g., prospective deployment in a hospital workflow) is generally considered outside the scope of ICLR, which focuses on machine learning methodology. Our evaluation framework aligns with the standard protocols established by recent foundational medical VLM works accepted at ICLR and similar venues (e.g., CARZero, MAVL, MedKLIP, GLoRIA, MGCA). Like these works, we employ retrospective evaluation on diverse, multi-source benchmarks to demonstrate methodological advancements. We go beyond standard metrics by including stress tests on unseen diseases, rare classes, and cross-modality transfer.
> >
> > We hope this clarifies that our evaluation is both rigorous and aligned with the field's standards. Given the feedback from the other reviewers, recognizing the value of our robustness contributions and extensive benchmarking, we respectfully ask the reviewer to reconsider the evaluation. Thank you!

---

> > > ### Comment · Reviewer_b7Jv · 2025-11-27
> > > **Final Reviewer Response**
> > >
> > > Thank you for the rebuttal. While I appreciate the clarifications and additional analysis, my core concerns remain.
> > >
> > > The methodological contribution still appears primarily engineering-oriented, and the rebuttal does not substantially change my assessment regarding novelty.
> > >
> > > First, the phenomenon described as “realistic clinical prompt variations such as synonyms or jargon” has already been studied as an open-vocabulary issue in clinical description [1].
> > >
> > > Second, the claimed novelty around “prompt sensitivity” has been extensively examined in prior work on prompt robustness and sensitivity, including [2, 3]. Therefore, this is not a new insight introduced by this paper.
> > >
> > > In addition, both the abstract and conclusion continue to claim that the proposed approach leads to “clinically reliable” radiology-facing VLMs. However, no real-world or clinical evaluation is provided to support such a conclusion.
> > >
> > > I also note that the authors stated in the rebuttal that clinical deployment or real-world evaluation is “outside the scope” of the work. This confirms that the current evidence does not justify the clinical reliability claims made in the manuscript.
> > >
> > > For these reasons, my evaluation remains unchanged.
> > >
> > > ---
> > >
> > > References
> > >
> > > [1] Towards Universal Text-driven CT Image Segmentation
> > >
> > > [2] Benchmarking Prompt Sensitivity in Large Language Models
> > >
> > > [3] Quantifying Language Models’ Sensitivity to Spurious Features in Prompt Design

---

> > > > ### Author Response · Authors · 2025-11-27
> > > >
> > > > We also appreciate the reviewer for pointing out the related work. Those work will be formally cited and discussed in our final draft.

---

> ### Author Response · Authors · 2025-11-27
>
> We thank the reviewer for the follow-up comments and the opportunity to clarify several remaining points.
>
> **W1(novelty and relation to prior work)**
>
> The cited works: Towards Universal Text-driven CT Image Segmentation [1], Benchmarking Prompt Sensitivity [2], and Quantifying Sensitivity to Spurious Features [3], indeed mention the open-vocabulary recognition and prompt robustness. However, these works mainly characterize the existence of prompt sensitivity or open-vocabulary issues, but do not introduce a similar strategy tailored to medical VLMs. The absence of such a solution in prior literature actually reinforces the value of our method: our approach is intentionally simple and efficient, yet directly addresses an open and unsolved practical limitation documented by the reviewer’s own citations.
>
> **W2 (“clinical reliability” and evaluation protocol)**
>
> Our usage of “clinically reliable” follows the convention commonly adopted in medical-AI benchmarking papers, which refers to robustness and consistency under clinically realistic variations rather than real-world hospital deployment.
>
> Importantly, our evaluation protocol follows the same domain standard as the reviewer-cited paper "Towards Universal Text-driven CT Image Segmentation" (from Medical AI top-tier journal). That work also writes:
>
> “…performance often declines when applied to diverse, real-world clinical data.”
>
> “Current text-prompt models… struggle to process the complex and diverse scenarios of real-world clinical applications.”
>
> Yet the evaluation is exclusively conducted on publicly available datasets annotated by clinical experts, without any in-hospital deployment. Our study follows precisely this established and widely accepted benchmarking paradigm.
>
> Thus, our empirical setup is aligned with field norms: using clinically annotated public datasets to evaluate whether a model behaves robustly under clinically plausible prompt variations. This type of validation is the standard and appropriate evidence for research-stage medical VLM work.
>
> **W3 (Clarification on scope)**
>
> Our claims are based on our observations. We dont claim deploy-ready level, but an enhanced clinical reliability that targets potential future translation of the technology. And we already update the draft with "results suggesting reliability" as your advice.
>
> We note that we has experience in deploying FDA-certified AI solutions and our current efforts in this domain is in the preparation of the ethics documents needed to start a single-arm clinical evaluation
>
> We hope these clarifications resolve the concerns regarding novelty and evaluation protocol.

---

### Official Review · Reviewer_1Rj5 · 2025-10-27

**Soundness:** 3
**Presentation:** 3
**Contribution:** 3
**Rating:** 6
**Confidence:** 5

**Summary:**

The study introduces **KEPIL** (Knowledge-Enhanced Prompt Image Learning), a framework designed to improve VLMs for medical disease detection in radiology. The paper includes 2 main contributions:  **Knowledge-Grounded Prompt Enrichment**, and  **Semantic-Aware Contrastive Loss**. The proposed loss function make the model robust to prompt variations. It uses a dual-embedding objective to align the embedding of equivalent prompt variants, teaching the model that different phrasings of the same concept are related. **KEPIL** achieved SOTA performance in zero-shot setting classification. It demonstrated robustness to prompt variations.

**Strengths:**

1) The proposed method re-defines prompt sensitivity problem in medical VLMs as a knowledge  alignment problem, aligning ontology-grounded text with visual features to stabilize predictions.

2) Demonstrating robustness to real prompt noise, improving performance on rare/unseen diseases, indicates meaningful impact  in clinical setting.

2) The study demonstrates consistent in zero-shot settings and smaller drops under prompt perturbations then competing baselines.

**Weaknesses:**

1) I have one concern about the improvement from the vision encoder being pretrained on chest X-ray data. The current gains might partially reflect this pretraining rather than the proposed knowledge components.

2) I am very confused because the experiment setting section mentions SIIM-ACR for the task of segmentation, but I could not find relevant report for this dataset in segmentation setting.

3) The study generates prompt variants including rephrasing, typos, omissions, and incorrect punctuation. However, it is not enough for realistic clinical settings such as abbreviations, multilingual terms, or clinician-specific jargon.

4) The study's introduction emphasis on long-tail distribution, but the results are not depicted the results for rare diseases. The results mainly focus on CXR diseases.

**Questions:**

1) Could you provide experimental results to clarify the concern (1) in the **weakness** section.

2) Could you provide experimental results on a rare disease dataset to support the paper claims?

3) Could you provide experimental results to clarify the concern (3) in the **weakness** section.

4) Could you provide additional subsection to explain the results for the segmentation task?

5) How does the Knowledge Query Module (KQM) enhance localization compared to baseline?

---

> ### Author Response · Authors · 2025-11-19
>
> # Response to Reviewer 1Rj5
>
> We sincerely thank the reviewer for the thoughtful summary and the positive assessment of our work’s soundness, presentation, and contribution. We greatly appreciate the reviewer’s clear understanding of our motivations and the recognition of our efforts in redefining prompt robustness through knowledge-grounded alignment. We are also grateful for the insightful concerns regarding pretraining effects, segmentation reporting, clinical prompt realism, and rare-disease evaluation. These points are highly constructive and have helped us improve the clarity and completeness of the work. We address each question and weakness in detail below.
>
> ### Q1W1 (Image encoder)
>
> All experiments in the ablation study (Table 4) were based on the **same backbone and same checkpoint** (identical to CARZero) pretrained on MIMIC-CXR, demonstrating that KEPIL’s effectiveness stems from its unique prompt enhancement and loss design.
>
> ### Q2W4 (rare disease dataset)
>
> We clarify that the performance of KEPIL on rare diseases was already presented in **Table 2 (PadChest-rare)** of the original submission, where KEPIL achieved the best performance.
>
> ### Q3W3 (realistic clinical settings)
>
> Thanks for the suggestion. We further evaluate the robustness of KEPIL against four clinically common prompt variations that frequently occur in real-world deployment:
> (1) abbreviations,
> (2) multilingual prompts (Spanish and French),
> (3) clinicians’ jargon, and
> (4) medical synonyms
> (all generated prompts will be added to the supplementary materials).
>
> **Table: Robustness evaluation against four clinically common prompt variations on CheXpert and ChestX-ray14 test sets (macro-averaged AUC, higher is better)**
>
> | Model            | CheXpert          |                 |                 |                 | ChestX-ray14      |                 |                 |                 |
> |------------------|-------------------|-----------------|-----------------|-----------------|-------------------|-----------------|-----------------|-----------------|
> |                  | Abbr. | Multi-lingual | Jargon | Synonym | Abbr. | Multi-lingual | Jargon | Synonym |
> | CarZero          | 78.81 | 62.07         | 79.85  | 85.21   | 69.86 | 65.75         | 71.85  | 73.40   |
> | **KEPIL (Ours)** | **89.54** | **86.03** | **89.89** | **90.62** | **79.56** | **76.73** | **78.98** | **79.56** |
>
> CarZero suffers severe performance degradation, with the most dramatic drop from **92.38 → 62.07 AUC** under multilingual prompts on CheXpert. In contrast, KEPIL maintains consistently high performance across all variants and both datasets, achieving only a minor drop (from **91.21 → 86.03**). These results demonstrate that our knowledge-enhanced prompt invariance learning effectively mitigates the sensitivity of medical VLMs to natural linguistic variations encountered in clinical practice.
>
> ### Q4 (segmentation result)
>
> Thank you for your suggestion. We have added a more detailed segmentation results analysis in the updated draft.
>
> ### Q5 (KQM module)
>
> We clarify that the major contributor to the localization ability is the prompt enhancement itself, as shown in **Table 6** of the original submission.
>
> ### W2 (SIIM-ACR segmentation)
>
> Thank you for pointing it out. It has been corrected to **RSNA Pneumonia**. We have also added the new segmentation results related to the SIIM-ACR dataset in the revised draft.

---

> > ### Comment · Reviewer_1Rj5 · 2025-11-24
> >
> > Thank you very much for team for the quality revision.
> > The revision demonstrates that the proposed method achieved consistent performance across several tasks and datasets. Especially, it reduces the prompt-sensitive of LLM-generated descriptions without grounding. However, I still concern about the robustness of the proposed method because the experiments focus only related common chest diseases.

---

> > > ### Author Response · Authors · 2025-11-24
> > >
> > > Thank you for your positive feedback on our revision. We greatly appreciate your recognition of the proposed method's consistent performance across multiple tasks and datasets, as well as its ability to mitigate prompt sensitivity in LLM-generated descriptions.We also appreciate your comment regarding the need to further evaluate robustness on a broader spectrum of thoracic conditions.
> > >
> > > To address this concern, we performed additional zero-shot inference experiments on the **PadChest-Full** dataset, which contains **122** heterogeneous radiographic labels spanning common diseases, rare findings, congenital anomalies, surgical materials, degenerative changes, and device-related observations. This dataset substantially expands the evaluation scope beyond standard chest X-ray benchmarks by including diverse conditions such as post-surgical artifacts, chronic structural changes, masses/nodules, congenital variants, and a wide set of pleural, parenchymal, and mediastinal abnormalities.
> > >
> > > **Results on PadChest-Full:**
> > >
> > > | Method            | AUC (%) |
> > > |-------------------|---------|
> > > | **KEPIL (Ours)**  | **77.42** |
> > > | CARZero           | 76.21   |
> > > | KAD               | 73.50   |
> > >
> > > These findings demonstrate that **KEPIL maintains its advantages even under a larger and more diverse label space**, supporting the **robustness** and **generalizability** of our approach. We hope this additional evaluation fully addresses your concern, and we would be happy to provide further analysis if helpful.

---

> > > ### Author Response · Authors · 2025-11-24
> > >
> > > The full list of findings included in the new evaluation is as follows:
> > >
> > > [normal, pulmonary fibrosis, chronic changes, kyphosis, pseudonodule, ground glass pattern, unchanged, alveolar pattern, interstitial pattern, laminar atelectasis, pleural effusion, apical pleural thickening, suture material, sternotomy, endotracheal tube, infiltrates, heart insufficiency, hemidiaphragm elevation, superior mediastinal enlargement, aortic elongation, scoliosis, sclerotic bone lesion, supra aortic elongation, vertebral degenerative changes, goiter, COPD signs, air trapping, descendent aortic elongation, aortic atheromatosis, metal, hypoexpansion basal, abnormal foreign body, central venous catheter via subclavian vein, central venous catheter, vascular hilar enlargement, pacemaker, atelectasis, vertebral anterior compression, hiatal hernia, pneumonia, diaphragmatic eventration, consolidation, calcified densities, cardiomegaly, fibrotic band, tuberculosis sequelae, volume loss, bronchiectasis, single chamber device, emphysema, vertebral compression, bronchovascular markings, bullas, hilar congestion, exclude, axial hyperostosis, aortic button enlargement, calcified granuloma, clavicle fracture, pulmonary mass, dual chamber device, increased density, surgery neck, osteosynthesis material, costochondral junction hypertrophy, segmental atelectasis, costophrenic angle blunting, calcified pleural thickening, hyperinflated lung, callus rib fracture, pleural thickening, mediastinal mass, nipple shadow, surgery heart, pulmonary artery hypertension, central vascular redistribution, tuberculosis, nodule, cavitation, granuloma, osteopenia, lobar atelectasis, surgery breast, NSG tube, hilar enlargement, gynecomastia, atypical pneumonia, cervical rib, mediastinal enlargement, major fissure thickening, surgery, azygos lobe, adenopathy, miliary opacities, suboptimal study, dai, mediastinic lipomatosis, surgery lung, mammary prosthesis, humeral fracture, calcified adenopathy, reservoir central venous catheter, vascular redistribution, hypoexpansion, heart valve calcified, pleural mass, loculated pleural effusion, pectum carinatum, subacromial space narrowing, central venous catheter via jugular vein, vertebral fracture, osteoporosis, bone metastasis, lung metastasis, cyst, humeral prosthesis, artificial heart valve, mastectomy, pericardial effusion, lytic bone lesion, subcutaneous emphysema, pulmonary edema, flattened diaphragm, asbestosis signs, multiple nodules, prosthesis, pulmonary hypertension, soft tissue mass, tracheostomy tube, endoprosthesis, post radiotherapy changes, air bronchogram, pectum excavatum, calcified mediastinal adenopathy, central venous catheter via umbilical vein, thoracic cage deformation, obesity, tracheal shift, external foreign body, atelectasis basal, aortic endoprosthesis, rib fracture, calcified fibroadenoma, pneumothorax, reticulonodular interstitial pattern, reticular interstitial pattern, chest drain tube, minor fissure thickening, fissure thickening, hydropneumothorax, breast mass, blastic bone lesion, respiratory distress, azygoesophageal recess shift, ascendent aortic elongation, lung vascular paucity, kerley lines, electrical device, artificial mitral heart valve, artificial aortic heart valve, total atelectasis, non axial articular degenerative changes, pleural plaques, calcified pleural plaques, lymphangitis carcinomatosa, lepidic adenocarcinoma, mediastinal shift, ventriculoperitoneal drain tube, esophagic dilatation, dextrocardia, end on vessel, right sided aortic arch, Chilaiditi sign, aortic aneurysm, loculated fissural effusion, fracture, air fluid level, round atelectasis, mass, double J stent, pneumoperitoneo, abscess, pulmonary artery enlargement, bone cement, pneumomediastinum, catheter, surgery humeral, empyema, nephrostomy tube, sternoclavicular junction hypertrophy, pulmonary venous hypertension, gastrostomy tube, lipomatosis]

---

> > > > ### Comment · Reviewer_1Rj5 · 2025-11-25
> > > >
> > > > Thank you for your detailed response. I appreciate the additional and robustness ablation studies. They are addressed my concern. I would like to update my concern accordingly.

---

> > > > > ### Author Response · Authors · 2025-11-25
> > > > >
> > > > > Thank you very much for your thoughtful follow-up. We sincerely appreciate your feedback and are glad that the additional experiments have addressed your concern :).

---

### Official Review · Reviewer_JrKC · 2025-10-31

**Soundness:** 3
**Presentation:** 3
**Contribution:** 3
**Rating:** 4
**Confidence:** 3

**Summary:**

This paper proposes KEPIL (Knowledge-Enhanced Prompt–Image Learning) to mitigate two challenges of medical-imaging VLMs in zero-shot settings: prompt sensitivity and lack of external domain knowledge. The method has three key components: (i) ontology-constrained prompt expansion and standardization using external medical knowledge bases (e.g., UMLS, Radiopaedia) with LLM assistance; (ii) a semantics-aware contrastive loss that enforces representation consistency across text-side views via a lightweight adapter with dropout; and (iii) entity-centric report normalization using RadGraph to reduce free-text noise. The architecture uses a frozen clinical text encoder (e.g., BioClinicalMPBERT) with a trainable adapter, a ViT-B/16 visual encoder, and a Knowledge Query Module (KQM) for token-level cross-attention alignment between image patches and text tokens. Experiments cover seven chest X-ray benchmarks (classification/segmentation/localization), including seen/unseen/rare categories and cross-modality transfer (CXR→CT). KEPIL outperforms or matches strong baselines and exhibits reduced performance degradation under diverse prompt perturbations; ablations attribute gains primarily to knowledge enrichment and the proposed loss.

**Strengths:**

- Strong empirics across seen/unseen/rare and cross-modality settings; consistent zero-/few-shot gains, including with limited labels for segmentation.
- Robustness focus is carefully evaluated with multi-source LLM-generated and perturbed prompts; UMAP suggests tighter intra-class clusters.
- Interpretability: entity-centric text and Radiopaedia cues produce attention maps that align with clinical findings.

**Weaknesses:**

- Train–test gap in semantic alignment: The loss aligns two stochastic views of the same text, not explicit cross-variant positives (paraphrases, synonym mappings). Training with explicit variant pairs would better match the robustness claim.
- Theory is light: “Semantic-aware” is broad; a perspective via invariance subspaces, information bottleneck, or generalization bounds maybe would strengthen the conceptual grounding.

**Questions:**

- What is the size of the entity set E and its coverage per disease category? How are conflicts between UMLS and Radiopaedia resolved; what fraction is human-audited?
- Did you train with explicit paraphrase/synonym/noisy variant pairs as positives? If not, can you add such a loss and report gains vs. the current two-view objective? I think by incorporating cross-variant positive pairs during the training phase could further support the claim of being "variant-robust".
- At inference, do you require LLM calls to generate prompts, or rely on a pre-built normalized library?
- In robustness plots, are max token length and template structure matched across models?

---

> ### Author Response · Authors · 2025-11-19
>
> # Response to Reviewer JrKC
>
> We thank the reviewer for the clear summary, as well as the positive assessment of our work’s contributions and empirical strengths. We also appreciate the constructive points regarding semantic alignment, theory, and robustness evaluation, and we address each concern in detail below.
>
> ### Q1 (size of the entity set)
>
> Our entity set **E** contains **42 radiographic entities** extracted from RadGraph-parsed MIMIC-CXR reports, covering common observations, structural findings, devices, and morphological descriptors. When mapped to the 14 CheXpert/ChestX-ray14 categories, this vocabulary achieves **full coverage**, with every disease linked to multiple relevant entities. Although these entities do not directly overlap with rare or unseen categories in PadChest-Rare, PadChest-Unseen, or COVID-19, our knowledge-enhanced prompts leverage the visual cues represented by the 42 entities to support reasoning in such OOD cases. UMLS and Radiopaedia are compatible — UMLS provides a concise definition, while Radiopaedia supplies the “radiographic features” section. All knowledge augmentations used for pre-training have been audited.
>
> ### Q2W1 (explicit augmentation)
>
> We thank the reviewer for this insightful suggestion. To directly evaluate whether explicit paraphrase/synonym/noisy variant pairs would further enhance prompt-robustness, we conducted the requested ablation by incorporating an additional contrastive term that treats nlpaug library-generated variants (paraphrase/synonym/noisy with probability 0.1 separately) as hard positive pairs during training.
>
> **Table 1: Effect of adding explicit text augmentation (nlpaug) cross-variant positive pairs on top of our two-view dropout objective**
>
> | Setting                  | Dataset                          | AUC (%) |
> |--------------------------|----------------------------------|---------|
> | KEPIL (original)         | CheXpert (in-domain)             | 91.21   |
> | KEPIL (original)         | ChestX-ray14 (in-domain)         | **80.95** |
> | KEPIL (original)         | PadChest-Unseen (OOD)            | **79.05** |
> | KEPIL (original)         | COVID-19 CXR-2 (OOD)             | **79.55** |
> | KEPIL + nlpaug           | CheXpert (in-domain)             | **91.78** |
> | KEPIL + nlpaug           | ChestX-ray14 (in-domain)         | 80.87   |
> | KEPIL + nlpaug           | PadChest-Unseen (OOD)            | 77.28   |
> | KEPIL + nlpaug           | COVID-19 CXR-2 (OOD)             | 75.57   |
>
> **Detailed robustness on CheXpert under four clinical prompt variants**
>
> | Variant            | Abbreviation             | Multilingual             | Clinician jargon         | Medical synonym          |
> |--------------------|--------------------------|--------------------------|--------------------------|--------------------------|
> | KEPIL (original)   | **89.54**                | 86.03                    | **89.89**                | **90.62**                |
> | KEPIL + nlpaug     | 89.63                    | **87.84**                | 83.15                    | 89.87                    |
>
> For in-domain diseases, the two models perform similarly. However, incorporating explicit nlpaug-based positive pairs reduces OOD generalization (e.g., Δ = −3.98 AUC on COVID-19 CXR-2) and weakens clinical prompt robustness, particularly for clinician jargon (Δ = −6.7 AUC). The results show that while explicit augmentation helps slightly on multilingual prompts, the dropout-based version is substantially more robust on authentic radiological variants (jargon and medical synonyms). This pattern likely arises because general-domain augmentation tools introduce colloquial or structurally irregular phrasing that deviates from the radiology language register, creating unstable entity-central anchors. In contrast, our original two-view dropout objective stays strictly within the clinical language manifold and enforces invariance without distorting medical semantics, leading to higher robustness across all real clinical variants and better OOD performance. We thank the reviewer again as these results bring us interesting findings.
>
> ### Q3 (LLM call)
>
> During inference, **no online LLM calls are required**. All finding-specific prompts are generated offline beforehand, so the model can run efficiently with pre-constructed prompts.
>
> ### Q4 (max token)
>
> In the robustness plots, the maximum token length and template structures are well-matched across all models. We confirmed that no prompt exceeds the token limits, as all prompt transformations follow the official prompt formats used by the corresponding baselines.

---

> ### Author Response · Authors · 2025-11-19
>
> ### W2 (theory)
>
> We thank the reviewer for the suggestion. Our theoretical motivation follows the standard contrastive-invariance principle in contrastive learning: the goal is to enforce representation-level invariance across perturbations while preserving distinctions. We use dropout as the perturbation mechanism precisely because it stays on the original clinical language manifold, ensuring that both views remain semantically valid. In contrast, general-domain augmentations easily produce off-manifold text (i.e., colloquial or non-radiological phrasing).

---

> > ### Comment · Reviewer_JrKC · 2025-11-26
> >
> > Thank you for your detailed response. Most of my concerns have been addressed. I will update my evaluation accordingly.

---

> > > ### Author Response · Authors · 2025-11-26
> > >
> > > Thank you very much for your thoughtful feedback and for taking the time to review our rebuttal :). We sincerely appreciate your engagement and are grateful that our response has effectively addressed your concerns. We look forward to your updated evaluation and truly appreciate your consideration.

---

### Official Review · Reviewer_Vgdq · 2025-10-31

**Soundness:** 3
**Presentation:** 3
**Contribution:** 3
**Rating:** 4
**Confidence:** 4

**Summary:**

This paper introduces KEPIL, a knowledge-enhanced prompt-image framework designed to address the issues of prompt sensitivity and limited generalization in medical vision-language models. It incorporates a large amount of medical knowledge based on ontologies and leverages dynamic prompt enhancement guided by large language models (LLMs) to improve understanding. In addition, a semantics-aware contrastive loss is proposed to enhance prompt robustness, and entity-centered report standardization is employed to optimize information representation. Experiments on seven benchmark datasets demonstrate that KEPIL achieves state-of-the-art performance in zero-shot classification and segmentation tasks.

**Strengths:**

This work is the first to integrate medical ontological knowledge, dynamic prompt enhancement, and semantics-aware contrastive learning to improve prompt robustness in medical vision-language models. With comprehensive experiments across multiple datasets and tasks, it demonstrates clear methodology and achieves significant performance gains, highlighting its practical value for medical AI.

**Weaknesses:**

1.The LLM-based prompt enrichment lacks transparency and rigorous validation against raw knowledge sources, risking unquantified hallucinations.

2.The superior segmentation scores lack qualitative validation (e.g., mask visualizations), leaving the clinical precision of improvements unproven.

3.​​The complex inference-time prompts increase computational overhead, but efficiency (latency) is not benchmarked, hindering practicality assessment.

**Questions:**

1.Provide radiologist-evaluated proof that LLM-enriched prompts are more clinically valuable than raw knowledge-base text.

2.​​Was robustness tested beyond typos (e.g., clinical synonyms like "opacity" vs. "consolidation")?

​3.​Why was dropout chosen over explicit text augmentation for creating positive pairs in the contrastive loss?

​​4. Is performance on rare diseases due to unique feature learning or merely semantic proximity to common diseases in the knowledge graph?

If my main concerns are properly addressed, I would be willing to raise my evaluation.

---

> ### Author Response · Authors · 2025-11-19
>
> # Response to Reviewer Vgdq
>
> We sincerely thank the reviewer for the careful reading and for the positive evaluation of our work’s soundness, presentation, and contribution. We appreciate the recognition of our focus on improving prompt robustness in medical VLMs, as well as the constructive comments regarding transparency, qualitative validation, and efficiency. These suggestions are valuable, and we have addressed each concern with additional analysis and clarification in the rebuttal.
>
> ### Q1W1 (radiologist-evaluated proof)
>
> Thanks for the suggestion. We have conducted a double-blind experiment with a radiologist to evaluate the quality of GPT-4o augmented knowledge for diseases with (A) and without (B) available raw-text descriptions from Radiopaedia.
>
> **(A) With raw-text (format standardization only):**
> We only use the LLM for format standardization without altering the underlying clinical content. We invited a radiologist to perform a double-blind clinical readability assessment on 10 randomly selected entities from our ontology [pleural effusion, atelectasis, consolidation, pneumonia, fracture, nodule, hyperinflate, collapse, emphysema, fibrosis]. The scoring rubric was:
>
> - **1** — unclear, disorganized, or difficult to interpret
> - **2** — somewhat readable but awkward or inconsistent
> - **3** — acceptable but could be smoother
> - **4** — clear and well-structured
> - **5** — highly readable, polished, and professional
>
> Readability scores:
>
> | Source     | Scores                        | Avg |
> |------------|-------------------------------|-----|
> | Raw text   | [3,5,5,5,5,4,3,4,5,3]          | 4.2 |
> | GPT-4o     | [5,5,5,2,3,5,5,5,4,5]          | 4.4 |
>
> This indicates that LLM-based standardization overall improves clinical readability without modifying core medical content.
>
> **(B) Without raw-text (generation from scratch):**
> We note that in the supplementary materials section of the original submission, we compare the LLM-augmented knowledge with the Radiology ontology knowledge (Radiopedia - Experts generated), the GPT-4o achieves the best performance across metrics like F1-RadGraph. We further compared GPT-4o-generated descriptions with actual raw-text descriptions using the following expert accuracy rubric:
>
> - **1** — major factual errors or incorrect radiographic statements
> - **2** — multiple inaccuracies or misleading claims
> - **3** — mostly accurate with minor imprecision
> - **4** — accurate with only small refinements needed
> - **5** — fully accurate and radiologically precise
>
> Accuracy scores:
>
> | Source     | Scores                        | Avg |
> |------------|-------------------------------|-----|
> | Raw text   | [4,5,5,5,5,4,5,5,5,5]          | 4.8 |
> | from Radiopaedia) | | |
> | GPT-4o     | [5,5,5,4,2,5,5,5,3,5]          | 4.4 |
>
> As expected, GPT-4o is slightly less accurate than expert raw text (general-purpose model vs. specialized ontology). However, an average of 4.4 still lies between “accurate with small refinements needed” and “fully accurate,” confirming that the generated descriptions remain clinically usable.
>
> ### Q2 (robustness tested beyond typos)
>
> Thanks for the suggestion. We further evaluate the robustness of KEPIL against four clinically common prompt variations that frequently occur in real-world deployment: (1) abbreviations, (2) multilingual prompts (Spanish and French), (3) clinicians’ jargon, and (4) medical synonyms (all generated prompts will be added to the supplementary material).
>
> **Table 1: Robustness evaluation against four clinically common prompt variations on CheXpert and ChestX-ray14 test sets (macro-averaged AUC, higher is better)**
>
> | Model              | CheXpert Abbr. | CheXpert Multi-lingual | CheXpert Jargon | CheXpert Synonym | ChestX-ray14 Abbr. | ChestX-ray14 Multi-lingual | ChestX-ray14 Jargon | ChestX-ray14 Synonym |
> |---------------------|----------------|-------------------------|------------------|-------------------|---------------------|-----------------------------|----------------------|-----------------------|
> | CarZero             | 78.81          | 62.07                   | 79.85            | 85.21             | 69.86               | 65.75                       | 71.85                | 73.40                 |
> | **KEPIL (Ours)**    | **89.54**      | **86.03**               | **89.89**        | **90.62**         | **79.56**           | **76.73**                   | **78.98**            | **79.56**             |
>
> CarZero suffers severe degradation (most dramatically from 92.38 → 62.07 AUC under multilingual prompts on CheXpert), whereas KEPIL exhibits only minor drops and consistently outperforms CarZero across all variants.

---

> ### Author Response · Authors · 2025-11-19
>
> ### Q3 (dropout vs explicit text augmentation)
>
> We chose dropout-based perturbation over explicit text augmentation because it provides a controlled, embedding-level way to create semantically invariant positive pairs without introducing accidental meaning changes. To directly address the reviewer’s question, we conducted the requested ablation by adding an extra contrastive term that treats nlpaug-generated variants (paraphrase/synonym/noisy with p=0.1 each) as hard positive pairs.
>
> **Table 2: Effect of adding explicit nlpaug cross-variant positive pairs on top of our two-view dropout objective**
>
> | Setting                            | Dataset / Metric                              | AUC (%) |
> |------------------------------------|-----------------------------------------------|---------|
> | KEPIL (original)                   | CheXpert (in-domain)                          | 91.21   |
> | KEPIL (original)                   | ChestX-ray14 (in-domain)                      | **80.95** |
> | KEPIL (original)                   | PadChest-Unseen (OOD)                         | **79.05** |
> | KEPIL (original)                   | COVID-19 CXR-2 (OOD)                          | **79.55** |
> | KEPIL + nlpaug                     | CheXpert (in-domain)                          | **91.78** |
> | KEPIL + nlpaug                     | ChestX-ray14 (in-domain)                      | 80.87   |
> | KEPIL + nlpaug                     | PadChest-Unseen (OOD)                         | 77.28   |
> | KEPIL + nlpaug                     | COVID-19 CXR-2 (OOD)                          | 75.57   |
>
> **Detailed robustness on CheXpert under four clinical prompt variants**
>
> | Variant            | Abbreviation             | Multilingual             | Clinician jargon         | Medical synonym          |
> |--------------------|--------------------------|--------------------------|--------------------------|--------------------------|
> | KEPIL (original)   | **89.54**                | 86.03                    | **89.89**                | **90.62**                |
> | KEPIL + nlpaug     | 89.63                    | **87.84**                | 83.15                    | 89.87                    |
>
> While in-domain performance is comparable, adding explicit nlpaug pairs hurts OOD generalization (e.g., Δ = −3.98 AUC on COVID-19 CXR-2) and substantially weakens robustness on authentic clinical variants (especially jargon, Δ = −6.7 AUC). General-domain augmentation injects colloquial/irregular phrasing that deviates from the radiology register, creating unstable anchors. In contrast, our dropout-only objective remains strictly within the clinical language manifold, yielding superior real-world robustness.
>
> ### Q4 (source of performance gain)
>
> The improvement on unseen diseases does not stem solely from the pretrained text encoder. For COVID-19 (unseen during pretraining), using only the disease name yields merely 61.26 AUC, whereas adding our ontology-grounded definition + radiographic features boosts performance to 79.55 AUC, demonstrating that KEPIL effectively learns to leverage enriched clinical knowledge.
>
> ### W2 (qualitative results)
>
> We already provide zero-shot grounding/segmentation qualitative results in Figure 6 of the main paper. We will add fine-tuned segmentation examples in the final version.
>
> ### W3 (Latency benchmark)
>
> KEPIL uses offline LLM-based knowledge enhancement; all descriptions are pre-generated and are **not** used at inference time. Test-time latency therefore remains unaffected and competitive:
>
> **Table 3: Inference time (seconds per image)**
>
> | Method         | Time (s) |
> |----------------|----------|
> | CARZero        | 0.0917   |
> | KAD            | 0.0381   |
> | MedKLIP        | 0.0369   |
> | MAVL           | 0.0415   |
> | **KEPIL (Ours)** | 0.0386 |
>
> We again thank the reviewer for the insightful questions that helped strengthen the paper.

---

> > ### Comment · Reviewer_Vgdq · 2025-11-25
> >
> > Thank you for your detailed response. I appreciate the additional robustness experiments, ablation studies, and expert evaluation results you've provided. These efforts have addressed most of my concerns and clarified several points. Overall, I'm pleased with the progress, and I’ll be updating my evaluation accordingly.
> >
> > I’d like to mention that some of the text in the figures and tables is quite small, but titles are quite long. If possible, updating these elements would significantly improve the overall presentation of the paper.

---

> > > ### Author Response · Authors · 2025-11-25
> > >
> > > Thank you very much for your positive feedback and for updating your evaluation :).
> > >
> > > We appreciate your suggestion regarding the text size in the figures and the length of the titles. We will address them in our final draft.
> > >
> > > Thank you again for the constructive comments as they are very helpful for improving the overall quality of the paper.

---

### Author Response · Authors · 2025-12-02
**Rebuttal Summary**

We are grateful for the constructive feedback provided during the review process. We are pleased that our detailed clarifications, additional analyses, prompt (used in pretraining stage) quality assesment with Radiologists and extended robustness evaluations have **successfully addressed the concerns raised by Reviewers Vgdq, JrKC, and 1Rj5**, and we sincerely appreciate **their commitment to update their evaluations positively**. Their comments have helped further improve the clarity and completeness of the manuscript.

Regarding the comments from Reviewer b7Jy, we respectfully note that several of the concerns raised do not appear to be grounded in the actual content of our submission, nor are they supported by citations or domain-relevant methodological standards. For instance, the reviewer questions the novelty of our method but **does not provide any references or prior works demonstrating a similar approach**. In addition, the reviewer argues that our mention of clinical applicability is an overclaim because we did not conduct real clinical deployment. We clarify in the manuscript and rebuttal, clinical applicability in this context refers to evaluation on real clinical datasets, which we perform extensively across multiple large-scale public clinical datasets, rather than real clinical deployment, which is outside the standard scope of ICLR submissions. What's more,  **the reviewer’s own cited medical VLM papers likewise discuss clinical applicability without performing clinical deployment.**  This further confirms that our experimental design and claims adhere to the commonly accepted protocols of the domain.

We confirm that our experiments, benchmarking design, robustness evaluations, and multi-dataset analyses strictly follow the established practices of the medical VLM community, including the evaluation protocols adopted by foundational medical VLM works at ICLR and related venues. We therefore believe that these concerns arise primarily from misinterpretation rather than substantive technical issues.

Overall, we appreciate the opportunity to engage in this rebuttal process. We are encouraged by the positive feedback from multiple reviewers, the acknowledgment of the contribution’s significance, and the updated scores. We hope the PC/ACs will take into consideration **the broad consensus regarding the novelty,  technical correctness, clarity, and impact of our work.** Finally, we would give a super thanks to AC/PCs for your hard working and taking more responsibility than previous years in the current review stage.

---

### Meta-Review · Area_Chair_GcyC · 2025-12-23

**Summary:**

After multiple rounds of discussion, the authors have actively addressed most of the reviewers’ comments. The concerns raised by Vgdq, JrKC, and 1Rj5 have been largely resolved to the reviewers’ satisfaction. However, the major concerns from b7Jy remain unaddressed, including: 1-Lack of statistical significance in the experimental results — for example, evaluation on only 10 cases does not provide sufficient statistical validity. 2-The issue of prompt sensitivity is not a new observation; the paper lacks a thorough literature review on this topic, as well as comparison and analysis with prior work. 3-Over‑claiming “clinically reliable” without rigorous experimental evidence to substantiate the claim. In addition, the paper proposes an LLM‑enhanced strategy to address prompt sensitivity, which is a common limitation in many vision‑language models (VLMs). It is essential to demonstrate the generalizability of this approach across other VLM baselines, as this would be critical for positioning the value of the work. Given these unresolved concerns, the AC recommends rejection of the paper.

**Reviewer Concerns:**

The concerns raised by Vgdq, JrKC, and 1Rj5 have been largely addressed to the reviewers’ satisfaction, whereas the three major concerns raised by b7Jy remain insufficiently addressed.

**Reviewer Scores:**

Vgdq, JrKC, and 1Rj5 may reconsider their evaluations and change their scores, whereas b7Jy would not.

---

### Decision · Program_Chairs · 2026-01-26

Reject